# Does EGFR Signaling Mediate Orexin System Activity in Sleep Initiation?

**DOI:** 10.3390/ijms24119505

**Published:** 2023-05-30

**Authors:** Marina Kniazkina, Vyacheslav Dyachuk

**Affiliations:** A.V. Zhirmunsky National Scientific Center of Marine Biology, Far Eastern Branch, Russian Academy of Sciences, Vladivostok 690041, Russia; marigknyaz@gmail.com

**Keywords:** orexins, neurotransmitters, ErbB, zebrafish, sleep, EGFR

## Abstract

Sleep–wake cycle disorders are an important symptom of many neurological diseases, including Parkinson’s disease, Alzheimer’s disease, and multiple sclerosis. Circadian rhythms and sleep–wake cycles play a key role in maintaining the health of organisms. To date, these processes are still poorly understood and, therefore, need more detailed elucidation. The sleep process has been extensively studied in vertebrates, such as mammals and, to a lesser extent, in invertebrates. A complex, multi-step interaction of homeostatic processes and neurotransmitters provides the sleep–wake cycle. Many other regulatory molecules are also involved in the cycle regulation, but their functions remain largely unclear. One of these signaling systems is epidermal growth factor receptor (EGFR), which regulates the activity of neurons in the modulation of the sleep–wake cycle in vertebrates. We have evaluated the possible role of the EGFR signaling pathway in the molecular regulation of sleep. Understanding the molecular mechanisms that underlie sleep–wake regulation will provide critical insight into the fundamental regulatory functions of the brain. New findings of sleep-regulatory pathways may provide new drug targets and approaches for the treatment of sleep-related diseases.

## 1. Introduction

Sleep is an evolutionarily conserved process in a variety of animal taxa, including primarily mammals, birds, fish, and also invertebrates, such as insects and cephalopods [1,2]. The sleep process is provided by neurons active in sleep, which, when depolarized, suppress wakefulness circuits. The control of this system is provided by homeostatic mechanisms that determine the time of sleep. To date, almost nothing is known about the molecular mechanisms responsible for the transmission of information about the need for sleep to sleep-active neurons [3].

Sleep regulation has been studied in invertebrates and model vertebrates. Studies on mammalians have shown that cerebrospinal fluid contains sleep-inducing factors [4] and that tanycytes can sense and transport cerebrospinal fluid molecules to the regions of the brain that are involved in sleep regulation [5]. Moreover, it has been found that interactions between sleep-active neurons of the sleep and the waking center play a significant role in the initiation of non-rapid eye movement (NREM) sleep and in maintenance of both NREM sleep and rapid eye movement (REM) sleep [6]. Sleep-promoting neurons are inhibited by active transmitters like noradrenalin, acetylcholine, and 5-HT [7]. All of these facts are consistent with the assumption that there exist additional elements monitoring homeostatic signals in the cerebrospinal fluid that regulate sleep [4,8]. In turn, studies on invertebrates have shown that AVA neurons that act as a control of interneurons for reversal behavior are implicated in sleep regulation in *Caenorhabditis elegans. C. elegans* showed decreasing activity in the AVD and AVA neurons [9].

However, a number of intriguing moments may be indicated as requiring or promising for further and detailed study. Questions still remain as to why animals sleep, what are the mechanisms that initiate sleep, and how the regulation has evolved independently in invertebrates and vertebrates. Many things are already known about vertebrates [10,11,12]. Genetic studies have demonstrated that similar neural mechanisms control sleep in vertebrates and invertebrates. However, there is a lack of data in the literature to clarify what is the mechanism of sleep initiation and what controls the initial nervous processes leading to falling asleep in vertebrates and how [13]. The study of the mechanisms and regulation of sleep in humans is an important aspect of research aimed at maintaining human health. These studies can provide therapeutic insights into sleep disorders and various neurological and neuropsychiatric disorders such as Alzheimer’s disease (AD) [14,15,16,17]. Thus, the use of sleep mechanisms in invertebrates can be very productive in deciphering similar or more complex mechanisms in humans.

Many neurotransmitters, such as acetylcholine, melatonin, dopamine, adenosine, orexins, serotonin and gamma-aminobutyric acid (GABA), glutamate, and histamine, are known to be involved in the regulation of the sleep–wake cycle in all animals [10,11,18,19]. All these molecules and the systems using these molecules are involved in the regulation of sleep. However, as has been shown recently, the epidermal growth factor receptor (EGFR) system is also involved in sleep regulation in invertebrates [13]. In some of the systems listed, the interaction with the EGFR system has already been documented [20]. However, for most neuroregulatory systems, it is unclear whether the EGFR system is involved as an activity regulator. For instance, orexin systems are very important because they are responsible for the activity of all other day activity maintenance systems and sleep is impossible during their operation [10,11]. Is there any direct interaction between the orexin and EGFR systems, and how might they interact?

## 2. Epidermal Growth Factor Receptor

As is widely known, EGFR is actively involved in signaling and controlling cell proliferation. Changes in the gene structure of the receptors have been shown in cancer development. Currently, EGFR is actively used as a target for several therapeutic interventions against cancer [21]. Until recently, studies on the function of EGFR have been conducted only to assess its effect on cell proliferation. The *egfr* gene (also known as *erbb/erbb1* or *her/her1*) is a member of the receptor tyrosine kinase (RTK) superfamily, which includes other members, such as *erbb2/neu/her2*, *erbb3/her3*, and *erbb4/her4* [22,23]. EGFR was the first discovered member of the family [24]. EGFRs are known to activate high-affinity receptor ligands [25], such as epidermal growth factor (EGF), transforming growth factor-α (TGF-α), heparin-binding EGF (HB-EGF), and cellulin, as well as low-affinity ligands such as amphiregulin, epiregulin, and epigen [26] (Figure 1). There are also neuregulins (neuregulins 1–4), which are associated only with ErbB3 and ErbB4, while it is unknown what ErbB2 is associated with [27]. Proteins of this family originate from the precursors of integral membrane proteins, which become their soluble forms after cleavage and contain a three-loop structure known as the EGF-like domain [28,29,30] (Figure 1).

Like all RTKs, EGFR consists of an extracellular ligand-binding domain, a transmembrane domain, and a cytoplasmic domain that contains a conserved protein tyrosine kinase core (Figure 1). Following ligand binding, EGFR undergoes homo- or heterodimerization. The dimer of EGFR and ErbB2 results in the formation of the most effective receptor, which increases the binding to EGF [31,32]. The activation of the internal kinase domain after ligand binding causes phosphorylation of tyrosine residues in the cytoplasmic part, which become associating sites for adapter proteins with Src homology 2 domains. The main pathways for the activation of ligand binding to ErbB receptors are the Ras-Raf-MEK-ERK1/2, STAT3 and STAT5, and PI3K-Akt-mTOR cascades, which play fundamental roles in regulating cell differentiation, proliferation, and survival [23,33,34,35]. To date, all the components and functions of the EGFR pathway are not fully known. For the invertebrates *C. elegans* and *Drosophila melanogaster*, EGFR has been shown to control proliferation of several cell types [36]. In vertebrates, it is involved in the development and morphogenesis of organs [37]. Moreover, EGFR signaling pathway is potentially present in almost all animal taxa, with the exception of placozoans and cnidarians [38]. The first animal in the evolutionary tree known to have the EGFR system is the planarian *Schmidtea mediterranea* (Platyhelminthes) [38]. The components of the EGFR signaling pathway consistently change from phylum to phylum. The synthesis of all known EGF proteins occurs as the synthesis of transmembrane precursors that have an EGF-like domain [39]. In turn, *C. elegans* has only one ligand (lin-3), unlike vertebrates [40,41]. Only one receptor was found both in *C. elegans* and in *D. melanogaster* (let-23 and der, respectively) [42,43]. Vertebrates have all the classes of receptors described above [33]. The EGFR pathway has been shown to be involved in growth and healing processes, as well as in sleep control [38].

## 3. Regulation of Sleep by EGFR Signaling in Invertebrates

It is known that peptidergic signaling pathways regulate sleep in invertebrates. Previously, the EGFR pathway was mainly investigated in relation to organisms’ development, but the role of EGFR signaling in the regulation of certain behaviors was also recognized [14,44,45,46,47,48,49]. Genetic studies on *D. melanogaster* and *C. elegans* have shown that EGFR signaling is necessary for the normal sleep process in these invertebrates [14,44,46,48,49,50,51,52]. A genome-wide association study using a group of wild-caught inbred strains of *D. melanogaster* has revealed that EGFR signaling involves controlling the sleep process during long and short sleep durations [53].

In *C. elegans*, EGFR signaling genes are expressed in RIS, which is activated during sleep. Moreover, there are ALA neurons that cause sleepiness and activate RIS. EGFR signaling can depolarize RIS through the activation of sleep-promoting neuropeptides, including those encoded by the neuropeptide genes flp-13, nlp-8, and flp-24 [13]. The hypothesis has been put forth that the peptides encoded by these genes can modulate sleep processes and suppress certain behaviors, including feeding [14,51,52]. Thus, the function of EGF during sleep is to promote drowsiness and participate in sleep modulation [13].

In *D. melanogaster*, the EGFR signaling pathway that induces cellular stress and mediation is suppressed by a mutation in the neuropeptide RFamide [54]. In other studies involving gain- and loss-of-function, EGFR signaling was involved in the maintenance of normal sleep processes in *C. elegans* and *D. melanogaster* [14,46,48,50]. In addition, sleep was reduced in zebrafish by damage to the genetic structure of *egfr*, *egf*, and *tgfa* [55]. These data indicate that variations in EGFR signaling genes determine the main characteristics of the sleep process, suggesting that increased functional activity leads to increased sleepiness. However, the effects of EGFR signaling on sleep require further investigation.

## 4. Regulation of Sleep by EGFR in Vertebrates

Vertebrates have a wide variety of pathways for EGFR and ligands [38,56]. The elevated levels of serum TGF-α, which is a mitogenic polypeptide and a member of the EGF family, are associated with fatigue in humans [57]. However, a study on rabbits treated with EGF injections showed that EGF was effective in enhancing sleep [44], although the effect was observed only when high doses of EGF were used. In contrast, a TGF-α infusion did not affect sleep duration in hamsters, although it suppressed daily activity [45,57]. Another evidence of the role of EGFR signaling activity in sleep processes in mammals was provided by the recent finding that extracellular signal-regulated kinase (ERK) slowdown leads to reduced sleep in rodents [58].

Recently, studies on the zebrafish *Danio rerio* have demonstrated the role of EGFR signaling in sleep processes. It has additionally been found that EGFR signaling is indispensable for the sleep recovery response to sleep deprivation. The mechanism of action of EGFR signaling in sleep has been described through mitogen-activated protein kinase/ERK and RFamide signaling. Furthermore, the dependent EGFR signaling and RFamide systems control sleep processes in *D. rerio* [55]. Overexpression of TGF-α increases the duration of sleep, while mutations in *EGFR* and genes encoding EGFR ligands or suppressing EGFR signaling decrease the sleep duration [55]. The authors hypothesize that decreased sleep duration due to the inhibition of EGFR signaling is caused by the disturbance in sleep processes.

The hindbrain and hypothalamus in the rodent and zebrafish brain are anatomically and molecularly conserved [2] and play a certain role in sleep control [59]. The main sleep regulation centers are located as described above. In rodents, EGFR is expressed in cells called tanycytes, whose long processes extend deep into the hypothalamus, the center of the sleep–wake cycle [8,60]. TGF-α is expressed in the ventricular system of the base ependymal cells and the central canal of the spinal cord [61,62]. In zebrafish, EGFR is also expressed in juxtaventricular cells, which express the glial cell marker SOX-2 and the glial fibrillary acidic protein (GFAP), while TGF-α is expressed in a population of SOX-2-expressing glial cells [55]. These are the observations of the conserved neuroanatomical structure of TGF-β/EGFR-regulated sleep. It has been found that the overexpression of TGF-α in zebrafish leads to the expression of c-fos in juxtaventricular cells expressing EGFR. However, an intracerebroventricular injection of either TGF-α or EGFR into hamsters reared in similar conditions led to the activation of ERK in juxtaventricular tanycytes expressing EGFR [60,63].

In vertebrate animals such as zebrafish, the EGFR signaling for sleep was regulate the expression of the pro-FMRFamide-related neuropeptide vasoactive intestinal peptide-like factor (VF) system or RFamide-related peptide precursor (NPVF) [55]. The RFamide neuropeptide family and NPVF-producing neurons are indispensable for sleep control in zebrafish [64]. The increasing level of expressions of the components of the EGFR signaling pathway regulates the NPVF expression, neuronal expression, and NPVF-expressing activity, which is a sensitive relationship between the two pathways. However, it is currently unknown how the EGFR signaling pathway is mediated by NPVF expression and the function of NPVF-producing neurons [55].

An analysis of sleep characteristics in European individuals from the British Biobank dataset has revealed a correlation between variations in the common genomic loci of kinase suppressor of Ras 2 (KSR2) and ErbB4 and sleep [64,65,66,67,68,69]. KSR2 is critical for EGFR signaling since it is a scaffold protein that mediates the interaction of the EGFR signaling components proto-oncogene serine/threonine-protein kinase (RAF), mitogen-activated protein kinase (MEK), and extracellular signal-regulated kinase (ERK) pathways [70]. ErbB4 is a member of the ErbB1/EGFR receptor tyrosine kinase family, which binds to several EGFR ligands and forms heterodimers with other members of the ErbB family, including EGFR [71]. EGFR signaling is an evolutionarily conserved control mechanism of the sleep system, which is involved in the development of human sleep disorders [55]. The mechanisms linking EGFR signaling to other sleep regulatory systems are yet to be elucidated by studying the role of EGFR signaling in sleep control systems in vertebrate model organisms and humans.

## 5. The Neuronal Cell System of Wakefulness Regulation in Invertebrates

In *D. melanogaster*, sleep is regulated through mushroom body neurons and dorsal paired medial neurons [1,72]. Neurons located in this area may increase activity if there is sleep deprivation. The hypothesis has been advanced that these neurons are responsible for the regulation of sleep processes [73,74,75]. Overexpression of TGF-α in *D. melanogaster* affects sleep duration and contributes to increased sleep time. However, inhibition of EGFR causes a reduction in sleep duration [46].

Neurons, referred to as RIS, are necessary to regulate the processes of sleep and falling asleep in *C. elegans* that occur through the activation of these neurons. EGFR signaling genes are expressed in RIS. Recently, it has been found that the EGFR signaling pathway activates RIS under cellular stress. Activation of ALA neurons responsible for day activity contributes to the shutdown of RIS neurons in *C. elegans* [13]. RIS neurons are GABAergic and peptidergic [76,77]. The ALA neurons regulate sleep duration by releasing neuropeptides [14,52]. The hypothesis exists that ALA neurons act in parallel with RIS neurons in regulating sleep duration [78,79]. The ALA neurons also perform the function of inhibiting the activity of other systems responsible for daily activity. However, the mechanism of ALA regulation of the EGFR system that initiates sleep is not clear [52,78].

## 6. The Wakefulness Regulation System in Vertebrates and the Orexin System

### 6.1. Orexins

Currently, the orexin (hypocretin)–melatonin interaction plays an important role in the molecular regulation of sleep–wake cycle. Neurons expressing orexins also play a significant role in the regulation of wakefulness in vertebrate animals [10,11]. The best-known mechanism underlying sleep disorders is the loss of orexin neurons [12].

Orexins 1 and 2, also known as orexins A and B, are two oligopeptides encoded by the same precursor gene [80,81]. Orexins are produced from a polypeptide precursor, proorexin, which is a fragment of the prepro-orexin protein (Figure 2). In the mammalian brain, prepro-orexin is expressed only in neurons confined to the lateral hypothalamus, which form a single cluster of cells [82,83]. However, orexin axons innervate other hypothalamic nuclei, the limbic system, thalamus, neocortex, trunk, and spinal cord. This type of organization of orexinergic neurons is conserved across all mammals investigated [84]. In addition to the regulation of sleep–wake cycle regulators, orexins are also involved in the regulation of the feeding behavior and cardiovascular system, energy homeostasis, arousal, and stress. Orexinergic neurons are projected, particularly to noradrenergic cells of the *locus coeruleus*, causing depolarization and activation of these cells. From the lateral hypothalamus, orexin bundles of axons to the spinal cord [66,83]. This fiber distribution has been proven through investigating the expressions of the orexin G protein-coupled receptors–orexin receptor 1 and orexin receptor [84,85] (Figure 2).

Orexin receptors 1 and 2 have different mRNA expression distributions, with some partial overlaps, suggesting that each receptor subtype plays different physiological roles. Orexin receptor 1 has been shown to be expressed in the prefrontal and infralimbic cortex, hippocampus, amygdala, bed nucleus of the *stria terminalis*, preoptic periventricular nucleus, anterior hypothalamus, dorsal raphe, ventral tegmental area, *locus coeruleus* nucleus, and laterodorsal tegmental nucleus/pedunculopontine nucleus in mammalian brain regions [86,87,88] (Figure 3). Orexin receptor 2 is expressed in the amygdala, tuberomammillary nucleus, hypothalamic arcuate nucleus, dorsomedial hypothalamic nucleus, paraventricular nucleus, lateral hypothalamic area, bed nucleus of the *stria terminalis*, paraventricular thalamus, dorsal raphe, ventral tegmental area, laterodorsal tegmental nucleus/pedunculopontine nucleus, CA3 of the hippocampus, and medial septal nucleus [87,88]. Studies using double in situ hybridization methods have shown colocalization of orexin mRNAs with other neurotransmitters in the tuberomammillary nuclei. Neurons exhibit orexin receptor 2 mRNA and vesicular monoamine transporter 2 (VMAT2) expressions and are positive for histamine, while orexin receptor 1 mRNA has not been detected in these neurons. In the dorsal raphe and median raphe nuclei, VMAT2-expressing serotonergic neurons are also positive to both orexin receptor 1 and 2 mRNAs. All VMAT2-positive noradrenergic neurons of the locus coeruleus show the co-expression of orexin receptor 1. In turn, orexin receptor 2 mRNA is detected inVMAT2-negative non-noradrenergic neurons. All vesicular acetylcholine transporter (VAChT)-positive neurons produce only orexin receptor 1 mRNA and are located in the pedunculopontine tegmentum and laterodorsal tegmentum. Orexin receptor 1-positive and/or orexin receptor 2-positive non-cholinergic neurons colocalize with cholinergic neurons [89]. In addition, orexins colocalize with the inhibitory peptide dynorphin [90,91], prolactin [92], pentraxin [93], and glutamate [94]. Orexin neurons are often glutamatergic [95,96], and the colocalization of orexins with gamma-aminobutyric acid-ergic (GABAergic) neurons has never been shown [95]. It has been shown, however, that orexin neurons promote glutamate release for histamine neurons in the tuberomammillary nucleus [97] (Figure 3).

The interaction of sleep-active neurons in the sleep–wake center of the preoptic area is suggested to play an important role in the initiation of non-rapid eye movement (NREM) sleep and maintenance of both NREM and rapid eye movement (REM) sleep [6]. During sleep, neurons of the ventrolateral preoptic area are activated and contain primarily GABA and galanin. Descending inhibitory projections are sent from these neurons to arousal-active neurons that express neurotransmitters of arousal, such as histamine, noradrenaline, 5-hydroxytryptamine (5-HT), and acetylcholine [6,98]. On the other hand, sleep-promoting neurons are inhibited by wakefulness-signaling molecules, such as noradrenaline, acetylcholine, and 5-HT [7]. The lack of orexin activity results in neurons being in the “go silent” state not only in the REM sleep phase but also in the period of wakefulness and is responsible for the onset of narcolepsy/cataplexy seizures [99]. The orexin system is involved in the regulation of waking/sleep states and the maintenance of wakefulness through the interaction of orexin neurons with monoaminergic/cholinergic nuclei in the brain. The orexins are linked to the limbic system, which controls emotional responses, the ventral tegmental reward system, and hypothalamic mechanisms which control food ingestion. In summary, orexins are responsible for perceiving the external and internal environment of the state of arousal [100].

Inhibition of the orexinergic system is mediated by the serotonin system in the raphe nucleus, the noradrenergic system in the colliculus through GABA neurons, nociceptive elements in the amygdala, the GABA/galanin system in the preoptic area, and the glycinergic system [101,102,103,104,105]. Increased levels of glucose and ghrelin in the blood also suppress the activity of orexinergic neurons [102,106], while histamine does not seem to affect orexin neurons [100]. Previous studies have suggested that 5-HT and noradrenergic neurons may provide inhibitory feedback to orexin neurons. Feedback mechanisms are suggested to control the activity of orexin and monoaminergic neurons. In turn, orexin neurons reverse the inhibitory effect of dopamine by acting on α2-adrenoceptors [107,108].

The orexinergic system is activated by the glutamatergic system, by neurons of the glutamatergic system that extend (bundles of) axons through the preceruleus and dorsomedial nucleus, as well as by the dopamine neurons of the ventral tegmentum. Innervation by orexin neurons and the presence of receptors for pineal orexins were observed in sheep [109], rats [110], and pigs [111], which demonstrated the conservation of the orexin regulation system in diurnal and nocturnal vertebrates. There is ample evidence that orexin neurons co-express glutamate [94]. More recently, glutamate release was also found in orexin cell bodies [112]. A study based on mutant mice showed that a large number of orexin neurons produce vesicular glutamate transporter-2 (VGluT2) [113]. Other transmitters activated by hunger, such as arginine-vasopressin, cerebral cholecystokinin, neurotensin, and oxytocin, also activate the orexin system [102,114]. The sulfated octapeptide forms of cholecystokinin, neurotensin, oxytocin, and vasopressin were shown to active orexin neurons [100,104,115,116]. Adenosine was also shown to inhibit orexin neurons through A1 receptors [117] (Figure 2).

One of the mechanisms of sleep–wake cycle control is the inhibitory effect of the melatonergic system on the orexinergic system. Neurons containing orexin and cells containing another neurotransmitter peptide, referred to as melanin-concentrating hormone (MCH), form overlapping projections, mainly in the lateral hypothalamus and *zona incerta*. In all vertebrates, melatonin is produced by the pineal gland in darkness and regulates diurnal and seasonal physiological changes [118,119]. In addition, MCH cells were shown in the reticular formation of the pons and caudal part of the laterodorsal tegmentum [120]. The MCH system inhibits orexin neurons. MCH neurons are active during REM sleep (REM-on-type), inactive during wakefulness, and weakly discharged in the non-REM sleep phase. It is hypothesized that the MCH system is responsible for the inhibition of awakening mechanisms and regulation of REM sleep [106,120].

Mammalian sleep is composed of a cycle beginning with NREM stage 1 that progresses through stage 2, then to stages 3 and 4, and ultimately culminates in REM sleep [121]. The measuring average length of the NREM-REM phases is one of the useful methods to analyze sleep quality. Several technologies exist for measuring sleep stage signals, for instance, EEG [122], EOG [123], acceleration [124], respiratory [125], and photoplethysmography [126]. The most informative of these methods is EEG because it shows that the alternation in brain waves follows the awake and sleep stages [127,128,129]. By interpretating signals during the transition from NREM to REM sleep, one can gain insight into various aspects of the sleep process, such as sleep efficiency in individuals with insomnia, prognosis of coma recovery, and the detection of drowsiness [130].

A study using data from patients with narcolepsy showed no differences in melatonin levels depending on the light cycles [131]. Humans display variable sensitivity to the hypnotic effects of exogenous melatonin, which causes small changes in the circadian clock [132]. However, there are no confirmed effects of melatonin-like hypnotics at night [133]. Moreover, it was suggested that tanycytes indirectly regulate feeding behavior and daily activity through the influence on the orexin system. The increase in DAGLα in tanycytes and its inhibition alters the expression of orexin neurons in response to the appearance of glucose [134].

Disruption of the orexinergic transmission has been observed not only in narcolepsy but also in AD, Parkinson’s disease (PD), and damaging brain injury. The characteristic symptoms of narcolepsy and PD are surplus daytime sleepiness and seizures of sleep. The loss of orexinergic neurons causes narcolepsy, in particular, type 1 narcolepsy [135]. In PD, the gradual destruction of orexinergic neurons occurs along with the destruction of nigrostriatal dopaminergic neurons or the dorsal striatum. Both processes disrupt the sleep–wake cycle [136]. Orexinergic neurons constitute a significant part of the system coordinating the modulation of the aminergic neurons of the brain, as they integrate incoming circadian signals, optical, and nutritional-metabolic activity [137].

### 6.2. Orexins and Other Regulatory Sleep Systems

The orexin system is highly conserved and has similar organizations in fish and mammals. Data from several fish species have confirmed that the orexin system regulates feeding behavior and physical activity in fish [138]. In the zebrafish brain, neurons containing orexin mRNA are located bilaterally only in the lateral rostral hypothalamus [139,140,141]. Most orexin fibers extend from the descending hypothalamic orexin neurons and pass through the ventromedial hypothalamus or posterior diencephalon, forming a ventral pathway that runs along the lateral edges of the ventral periventricular hypothalamus [139]. These processes innervate the posterior tuberal nucleus, caudal area of the periventricular hypothalamus, and suture area. The dorsolateral bundle innervates the nuclei of the thalamus [139]. The pretectal nuclei contain 5-HT and are densely innervated by orexin fibers. In addition, fibers contain orexins that innervate the optic nerve in the central gray layer, central landscape layer, lateral nuclei of the dorsal tegmentum, and tegmentum. The nuclei of the suture are densely innervated. The orexin fibers innervate the preoptic region and are immunoreactive to choline O-acetyltransferase (ChAT). Pretectal neurons that are immunoreactive to tyrosine hydroxylase (TH), 5-HT, or ChAT are also innervated by orexin fibers, which also innervate the anterior thalamic and tuberal catecholamine neurons [138]. The main area of distribution of orexin fibers is the raphe area, and the *locus coeruleus* also receives some contact from these fibers. Therefore, many aminergic and some cholinergic centers appear to have connections with orexin nerve fibers in fish (138]. Furthermore, the homology between fish and mammalian systems of organization of glutamatergic neurons relative to orexin neurons has already been shown [142,143] (Figure 3).

The melatonin system, as that in mammals, is a recognized sleep inducer in zebrafish (Zhdanova et al., 2001 [144]). In mammals, light activation is transmitted via the retinohypothalamic tract, and the signal travels through the central oscillator in the suprachiasmatic nucleus on pineal glands. The pineal glands produce melatonin by sympathetic innervation. The information about light/dark assay reaches the suprachiasmatic nuclei from retinal photosensitive ganglion cells of the eyes. The fish pineal gland is an autonomous organ capable of producing melatonin, and oscillators are located in the photoreceptor cells of the pineal gland [119,145]. There are only a few examples of pineal innervation in bony fishes [145] (Figure 3). However, another study shows orexinergic innervation in the pineal gland and orexin signals modulating pineal melatonin production [10]. Orexins in the zebrafish pineal gland stimulate the release of melatonin. The stimulation was shown by suppressing the *aanat2*—a gene that encodes the arylalkylamine N-acetyltransferase enzyme. It is an important regulator of the sleep–wake cycle (circadian rhythm) in the orexin receptor-mutant pineal glands of fish [146]. It is hypothesized that orexins contribute to sleep consolidation in the dark by stimulating the melatonin system to promote sleep. Without orexin signaling, the melatonin system and circuit may be activated to regulate the decrease in melatonin expression [10]. Moreover, the distribution of orexin fibers and mRNA strongly resembles that of melatonin receptor expression, particularly in the periventricular gray zone of the optic tissue, as well as in the periventricular thalamus and hypothalamus. The orexin and melatonin systems may also cooperate in brain regions where receptors of orexins and melatonin are produced [10]. The interaction between melatonin and the orexin pathway needs a comprehensive study to fully understand the regulatory mechanism.

The presence of studies suggests that the introduction of EGF-like peptides into the SCN in the hypothalamus can lead to changes in behavior, which may suggest a possible mutual regulation between the EGFR system and the orexin system. Additionally, existing research suggests that orexins may have interactions with other factors. Further investigation, such as disrupting the signaling of naturally occurring peptides, is required to gain a clearer understanding of the roles played by EGF-like peptides expressed in the SCN [147].

In zebrafish, the regulation of sleep–wake cycles, mediated by interactions between the orexin, melatonin, histaminergic, aminergic, and cholinergic regulatory systems, appears to be similar to that in mammals. More research is needed on the role of the orexin system in the sleep–wake cycle control to identify the molecular mechanism of interaction with receptors and the interaction of the orexin system with other systems.

## 7. Conclusions

This review summarizes information on the activity of the EGFR system in the regulation of sleep–wake cycles, the relationships between molecules in the sleep–wake regulation system, and the possible role of EGFR signaling in the molecular regulation of sleep. For invertebrates, the mechanisms of sleep regulation based on the EGFR signaling pathway and neurotransmitters together have been shown [13,38,51,52]. To date, the large array of data summarized has shown that the homeostatic modulation of sleep initiation processes exists in both invertebrates and vertebrate animals. However, the question as regards the pathways of the transmission of initiating signals to the neurons involved remains open. It has been pointed out that sleep is preceded by the deactivation of orexin neurons, with this process regulating the activity of other neurotransmitter systems during wakefulness and making them enter the functional “day activity” [99,100]. Inhibition of the orexinergic system affects the modulation of the aminergic neurons all over the brain [137]. The influence of the EGFR system on sleep regulation by neurotransmitter systems invertebrate release is expressed in RIS under cellular stress in *C. elegans.* In vertebrate animals, there exist prerequisites for the influence of the EGFR system on sleep regulation in rodents. The EGFR signaling activity in sleep processes was shown by the recent finding that ERK slowdown led to reduced sleep [58]. In *D. rerio*, EGFR signaling is indispensable for the sleep recovery response to sleep deprivation, and the dependent EGFR and RF neuroamide systems control sleep processes [55].

The EGFR system In invertebrates inhibits orexinergic activity and regulates the subordinated systems via the neurons that are active during sleep [13,38]. Of course, the EGFR signaling pathway may have one of its functions as a sleep regulator, not only for the orexin system. The EGFR system is assumed to be a higher modulation system in both groups of animals because, being the older in evolution, that system operates numerous processes—not only signaling and controlling proliferation, but also could be involved in the sleep regulatory system. Since sleep processes are regulated through the EGFR system as a higher modulation system, there should be some influence on the orexin system, as a system of daytime activation.

Based on the available data, one can assume that the EGFR signaling pathway plays an additional, still unknown role in the control of neuronal interactions both during sleep and in the nervous system processes in general. The EGFR system seems to regulate the interaction of groups of neurons responsible for the regulation of day brain activity. This confirms the assumption that there is a reverse regulation from the systems that regulate sleep through the EGFR system. Further, it could be a topic for investigating the mediating of the regulation of neuronal proliferation depending on the daily activity of the body through EGFR signaling.

Studies with animal models (such as *C. elegans*) have shown that the regulation of the EGFR likely leads to changes in neuronal activity during sleep. The question of how the EGFR signaling deactivation leads to sleep disturbance remains open. When a number of significant processes occur in the body (such as fatigue, falling asleep, plunging into darkness, hunger, etc.), the receptor systems from the sense organs must receive an appropriate signal leading to an adequate reaction of the body. The pathways of transmission of such signals for the initiation of sleep and activation of the orexin system are still unknown. We suggest that the main mechanism of transfer of signals to neurotransmitter systems is via the EGFR regulation system.

In turn, EGFR and the corresponding neurotransmitters perform their regulatory roles depending on the state of the body and continue to change the system in the absence of sleep. Further study is needed on the molecular interactions between the EGFR and neurotransmitter systems involved in the regulation of sleep processes in vertebrates. Studying conserved mechanisms is important not only from an evolutionary standpoint but also because these mechanisms are likely to modulate the regulation of sleep in all animals. The search for the possible line of succession from the invertebrate regulation system to the vertebrate one in evolution may be a promising field of research to clarify this interaction mechanism.

## Figures and Tables

**Figure 1 ijms-24-09505-f001:**
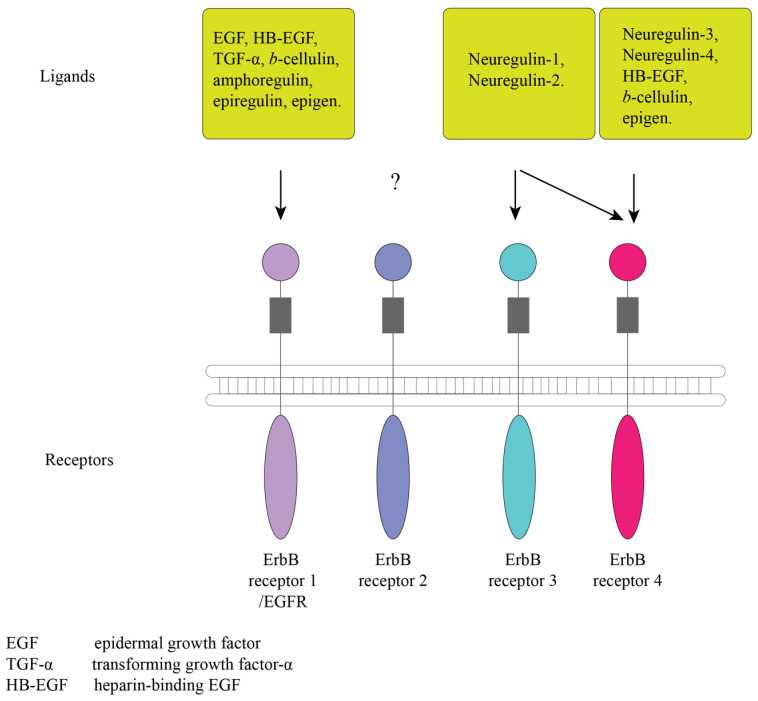
Representation of the ErbB receptors and ligands. The acronyms are as follows: EGF, epidermal growth factor; TGF-α, transforming growth factor-α; HB-EGF, heparin-binding EGF, ?—symbol indicates that it is unknown with what ErbB2 binds to [27].

**Figure 2 ijms-24-09505-f002:**
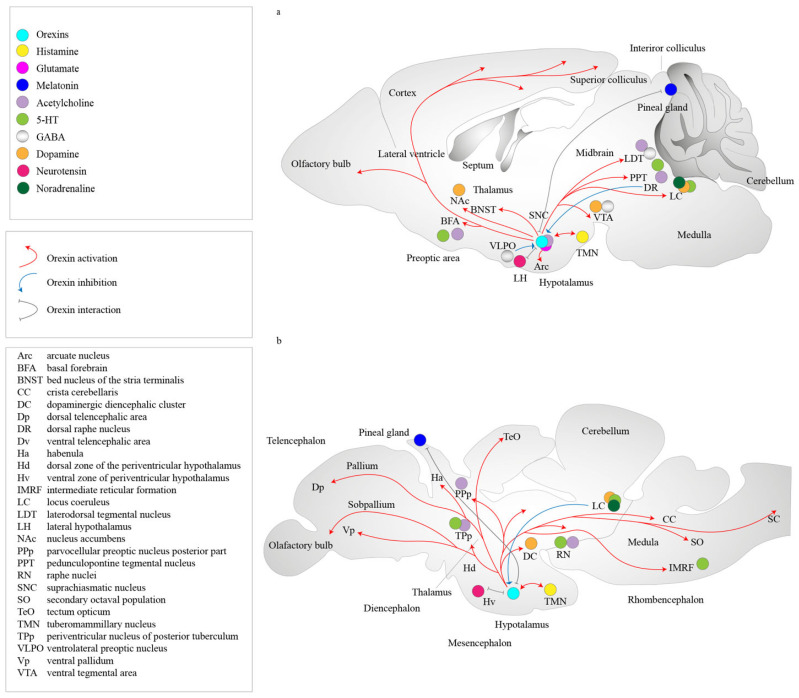
Representation of the metabolism of orexins and orexin receptors in the locus of the mammalian brain. Orexin A and orexin B are derived from a common precursor peptide, pre-proorexin. The actions of orexins are mediated via two G protein-coupled receptors, orexin receptor 1 and orexin receptor 2. Orexin receptor 1 shows a greater affinity for orexin A, whereas orexin receptor 2 is a non-selective receptor for both orexin A and orexin B. Cells in the TMN zone have orexin receptor 2, DR have orexin receptors 1 and 2, and LDT/PPT and LC have orexin receptor 1 cells in the brain, including loci DR, LC, LDT, PPT, and TMM. (**a**) Mammalian brain. (**b**) Fish brain. The abbreviations are as follows: 5-HT, 5-hydroxytryptamine; Arc, arcuate nucleus; BFA, basal forebrain; BNST, bed nucleus of stria terminalis; CC, crista cerebellaris; DC, dopaminergic diencephalic cluster; Dp, dorsal telencephalic area; DR, dorsal raphe nucleus; Dv, ventral telencephalic area; GABA, gamma-aminobutyric acid;Ha, habenula; Hd, dorsal zone of the periventricular hypothalamus; Hv, ventral zone of periventricular hypothalamus; IMRF, intermediate reticular formation; LC, locus coeruleus; LDT, laterodorsal tegmental nucleus; LH, lateral hypothalamus; MCH, melanin-concentrating hormone; Nac, nucleus accumbens; PPp, parvocellular preoptic nucleus posterior part; PPT, pedunculopontine tegmental nucleus; RN, raphe nuclei; SNC, suprachiasmatic nucleus; SO, secondary octaval population; TeO, tectum opticum; TH, tyrosine hydroxylase; TMN, tuberomammillary nucleus; TPp, periventricular nucleus of posterior tuberculum; Vp, ventral pallidum; VLPO, ventrolateral preoptic nucleus; VTA, ventral tegmental area.

**Figure 3 ijms-24-09505-f003:**
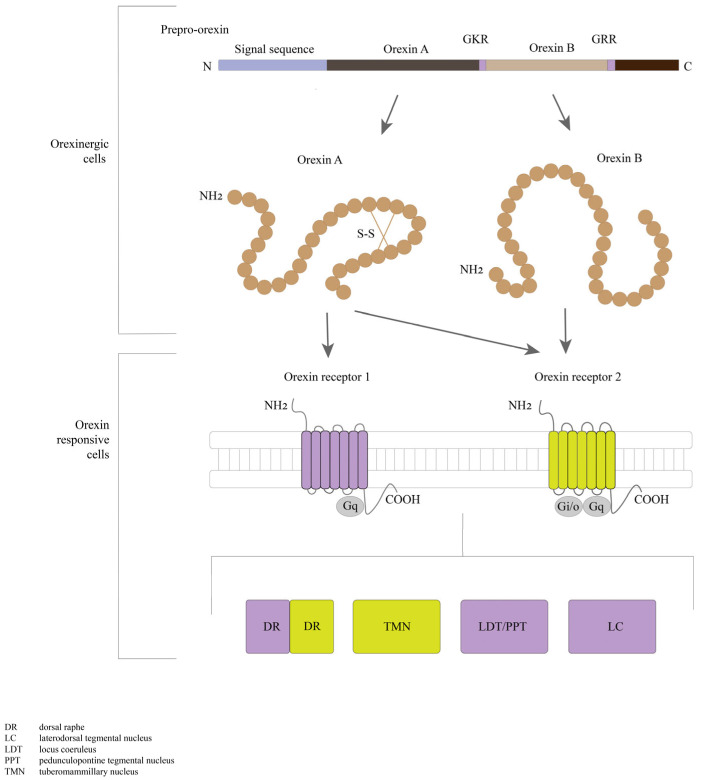
Schematic representation of interactions in the orexin system. Orexin neurons are found only in the lateral hypothalamic area but project throughout the central nervous system. Red arrows indicate excitatory projections, blue lines indicate inhibitory projections, and gray lines indicate the interactions. The acronyms are as follows: DR, dorsal raphe; LC, laterodorsal tegmental nucleus; LDT, laterodorsal tegmental nucleus; PPT, pedunculopontine tegmental nucleus; TMN, tuberomammillary nucleus.

## Data Availability

The data presented in this study are available on request from the corresponding author.

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
