# Peer review of "Does EGFR Signaling Mediate Orexin System Activity in Sleep Initiation?"

_ijms, 2023, doi:10.3390/ijms24119505_

Round 1

Reviewer 1 Report

Marina Kniazkina and Vyacheslav Dyachuk summarized in this review, the current status of the EGFR signaling pathway in the molecular regulation of sleep. The article gives an interesting historical and scientific perspective on the field.  It focuses mostly on EGFR signaling and orexin system.  Overall, it is an interesting paper, well organized and nicely written.

1.     The whole paper needs a quick copy edit. There are a few typos, and wrong verb tenses.

2.     In Section 6, it is better to dig deeper into the interaction between the orexin and EGFR systems, which is the topic of the review paper. With modifications this could be a very influential paper.

3.     Figure 2. is blurry. A high-resolution image is needed.

4.     As for References, I noticed that only about 13% references are primary research from the past 2-5 years.  

Author Response

Review1

 Comments and Suggestions for Authors

Marina Kniazkina and Vyacheslav Dyachuk summarized in this review, the current status of the EGFR signaling pathway in the molecular regulation of sleep. The article gives an interesting historical and scientific perspective on the field.  It focuses mostly on EGFR signaling and orexin system.  Overall, it is an interesting paper, well organized and nicely written.

1.The whole paper needs a quick copy edit. There are a few typos, and wrong verb tenses.

Reply

We have now gone through the manuscript thoroughly and corrected all such mistakes

2.In Section 6, it is better to dig deeper into the interaction between the orexin and EGFR systems, which is the topic of the review paper. With modifications this could be a very influential paper.

Reply

We now include an additional information about the interaction between the orexin and EGFR systems

3.Figure 2. is blurry. A high-resolution image is needed.

Reply

Apparently this happened because of the pdf file formation, which greatly compresses the figures. We have uploaded all Figs in tiff format and very high quality.

4.As for References, I noticed that only about 13% references are primary research from the past 2-5 years.

Reply

We have added "fresh" articles

Reviewer 2 Report

In this review, the authors have investigated recent studies regarding sleep-wake cycle. I have found the review quite clear, well structured, and informative.

The only issue for me is that the author should put more emphasis on the importance of sleep-wake cycle by giving exmaples. For instance, it is important to consider transition time from wakefulness to sleep stage I which is also known as sleep onset latency, e.g., 10.1109/JSEN.2022.3155345

Author Response

Review2

Comments and Suggestions for Authors

In this review, the authors have investigated recent studies regarding sleep-wake cycle. I have found the review quite clear, well structured, and informative.

The only issue for me is that the author should put more emphasis on the importance of sleep-wake cycle by giving exmaples. For instance, it is important to consider transition time from wakefulness to sleep stage I which is also known as sleep onset latency, e.g., 10.1109/JSEN.2022.3155345

Reply

Thank you for this suggestion, which we have followed. We have included the article proposed by the reviewers, added additional information about the importance of sleep-wake cycle, and interaction between the orexin and EGFR systems.